# Assessment of Seasonal Variability of Extreme Temperature in Mainland China under Climate Change

Weixiong Yan [1,2]⊙, Junfang Zhao [2,*], Jianping Li [1,*] and Yunxia Wang [1]

1   Key Laboratory for Meteorological Disaster Monitoring and Early Warning and Risk Management of Characteristic Agriculture in Arid Regions, Ningxia Key Laboratory for Meteorological Sciences, Ningxia Institute of Meteorological Sciences, Yinchuan 750003, China; laobencau@163.com (W.Y.); wyxia1995@nwafu.edu.cn (Y.W.)
2   State Key Laboratory of Severe Weather, Chinese Academy of Meteorological Sciences, Beijing 100081, China
*   Correspondence: zhaojf@cma.gov.cn (J.Z.); lijp_111@163.com (J.L.); Tel.: +86-130-0198-1203 (J.Z.); +86-138-9519-3106 (J.L.)

**Abstract:** Some studies have suggested that variations in the seasonal cycle of temperature and season onset could affect the efficiency in the use of radiation by plants, which would then affect yield. However, the study of the temporal variation in extreme climatic variables is not sufficient in China. Using seasonal trend analysis (STA), this article evaluates the distribution of extreme temperature seasonality trends in mainland China, describes the trends in the seasonal cycle, and detects changes in extreme temperature characterized by the number of hot days (HD) and frost days (FD), the frequency of warm days (TX90p), cold days (TX10p), warm nights (TN90p), and cold nights (TN10p). The results show a statistically significant positive trend in the annual average amplitudes of extreme temperatures. The amplitude and phase of the annual cycle experience less variation than that of the annual average amplitude for extreme temperatures. The phase of the annual cycle in maximum temperature mainly shows a significant negative trend, accounting for approximately 30% of the total area of China, which is distributed across the regions except for northeast and southwest. The amplitude of the annual cycle indicates that the minimum temperature underwent slightly greater variation than the maximum temperature, and its distribution has a spatial characteristic that is almost bounded by the 400 mm isohyet, increasing in the northwest and decreasing in the southeast. In terms of the extreme air temperature indices, HD, TX90p, and TN90p show an increasing trend, FD, TX10p, and TN10p show a decreasing trend. They are statistically significant ($p < 0.05$). This number of days also suggests that temperature has increased over mainland China in the past 42 years.

**Keywords:** STA; China; warm days; cold days; warm nights; cold nights; hot days; frost days





## 1. Introduction

The global surface temperature was 1.09 °C higher in last the last decade than 1850–1900, which was more likely not higher than for any multi-century average during the Holocene [1]. The frequency and intensity of hot extremes have increased and those of cold extremes have decreased on the global scale since 1950 [1]. Agriculture is one of the most directly affected sectors by global climate change, especially crop production and food security [2]. China is the largest food-producing country in the world. Since 2003, grain output has increased continuously. In 2020, the sown area of grain reached 1.17 million $\times 10^8$ hm$^2$ and a total yield of $6.69 \times 10^8$ t [3]. Chinese food production plays an important role in its own country and even food security in the world. China is located in eastern Eurasia, mostly at middle and high latitudes (Figure 1). This is a sensitive and significant area for global climate change. Climate change is likely to have a significant impact on global food production, and Chinese food production is also faced with the uncertainty caused by climate change and the risk of yield reduction caused by extreme climate [4–7]. In the past half-century, the yield of most major crops in the world has

increased significantly, mainly due to irrigation, chemical input, and the extensive use of modern crop varieties [8]. However, an increasing number of studies have shown that there are two significant differences between the positive and negative effects of climate warming on crop growth and yield [9–13], and the results depend on the study areas, crops, and methods.

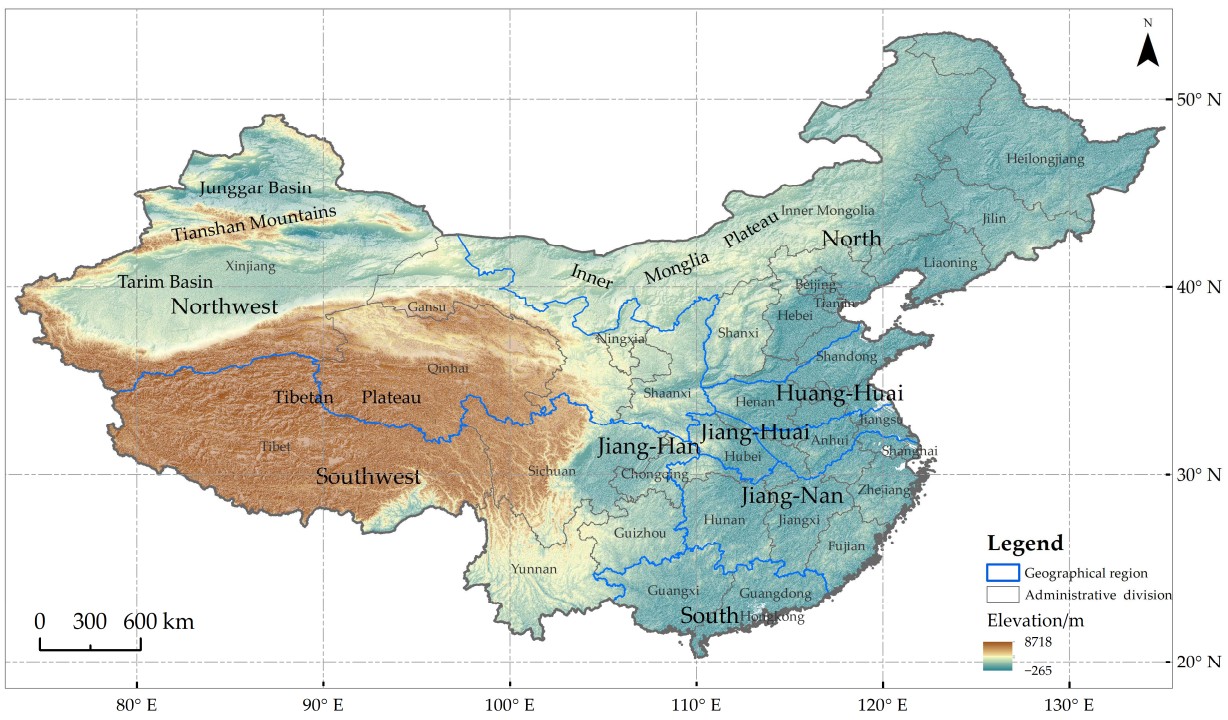

**Figure 1.** Map of mainland China including geographic divisions and provinces.

In the past 100 years, the trend of temperature increase in China has been higher than the global average [14]. In the last 50 years, surface air temperature has increased by nearly 1.40 °C with a change rate of 0.25 °C (10 years)$^{-1}$, indicating that the trend of climate warming has accelerated [14,15]. From 1961 to 2018, the start date of the average growing season in China advanced, the end date was delayed, the length was extended, and the advance of the start date had a greater impact on the length extension. In addition, the advance of the start date and the increase in the length of the average growing season in China are mainly due to warming in spring. The trend in the growing season length in China follows the Northern Hemisphere trend [16]. The extreme air temperatures also show an increasing trend. During 1961–2014, the temperatures of the hottest day and coldest night in China showed a rapid upward trend, which were 0.17 °C·10a$^{-1}$ and 0.52 °C·10a$^{-1}$, respectively [17]. From 1961 to 2018, cold days (TX10p) and cold nights (TN10p) in China had a decreasing trend, and the decreasing trend of TN10p was significantly greater than that of TX10p. Warm days (TX90) and warm nights (TN90) had an increasing trend, and the increasing trend of TN90p was significantly greater than that of TX90p [18]. The trend of extreme air temperature was consistent with that on the global scale [1]. Extreme high temperatures and low temperatures can damage crop tissues and organs, delay crop growth and development, or hinder flowering and fruiting, resulting in lower yields [19]. Long-term global warming has changed the distribution of temperature changes and extremely high temperatures have become more common in some places, such as southwest China [20–22]. The same is true for agricultural regions, where the probability of crop exposure to maximum temperatures increases at the critical stage of reproductive growth [23]. According to the latest studies, the contribution rate of climate warming to the yield of spring maize in northeast China from 1981 to 2009 was 29.7%, and maximum

temperatures above 30 °C caused a 14.1% yield reduction. The increase in high temperature during the vegetative period was the main reason for the yield reduction [24]. Many studies have also focused on the impact of temperature changes in different periods on crop yield. Some scholars have collected relevant literature worldwide and used meta-analysis to conclude that 0–5 °C warming during the reproductive period had significant negative effects on wheat yield and its components [25,26]. The decline in wheat yield is different in different climatic regions, and the negative effect of warming at night is greater than that during the daytime [25]. As far as China is concerned, wheat yield increases significantly in the monsoon region but decreases significantly in the temperate continental climate region. The winter wheat yield has increased significantly with the increase in night temperature in the monsoon region [26]. Another study indicated that warming up to +3 °C has increased winter yield by 5.8% per °C (change rate of yield/average of yield) while reducing spring wheat yield by 16.1% per °C [27].

The annual variability of temperature, precipitation, and plant phenology usually has seasonal cycles. With the growth of global climate studies, monitoring these seasonal trends as a means to detect the response of the Earth system to global change has sparked great interest [28–30]. Previous studies regarding the impact of climate warming on crop yield have more or less been related to the temporal dynamics of temperature [24–27]. Most existing studies have used long-series data to analyze annual and seasonal changes or the length of the growing season [14–18], but analysis of the seasonal trend of temperature is not sufficient, especially the changes in the time of occurrence for extreme air temperature. The purpose of this paper is to select several extreme air temperature indices from the ETCCDI (Expert Team on Climate Change Detection and Indices) [31], which are related to grain production. We analyze the spatial pattern and change in extreme air temperature seasonal trends in mainland China over the last 42 years and detect the changes in extreme temperature events. These results are expected to provide help for studying the long-term impact trend of climate change on food production.

## 2. Data and Methodology

### 2.1. Data

In recent years, with the development of automatic observation technology, the number of meteorological observation stations in China has increased greatly, which improves the spatial density and frequency of observations and partially meets the needs of land-atmosphere processes and weather climate analysis. However, at the beginning of the layout, automatic observation stations were mostly placed in sections with stable communication, convenient maintenance, and clear purpose, which have high relevance, but cannot provide uniform distribution and long-time series; therefore, it is difficult to meet the needs of long-term climate trend analysis at present. Reanalysis data are a set of gridded and long-series meteorological datasets based on data assimilation technology that integrate multisource observation data and numerical simulation results. This could compensate for the uneven spatial and temporal distribution of in-situ observations. At present, the main reanalysis datasets include a series of products (ERA5, ERA15, ERA40, and ERA-Interim) from the European Centre for Medium Term Weather Forecasts (ECMWF) [32], NCEP/NCAR reanalysis I (R1) jointly developed by National Centers for Environmental Prediction (NCEP) and National Center of Atmospheric Research (NCAR) [33], NCEP/DOE reanalysis II (R2) jointly developed by the Department of Energy (DOE) [34], Japanese 25 year reanalysis (JRA-25) and 55 year reanalysis (JRA-55) [35,36], NASA's Modern-Era Retrospective analysis for Research and Applications (MERRA and MERRA-2), etc. [37,38]. In May 2021, the China Meteorological Administration (CMA) released China's first-generation global/land surface reanalysis product (CRA). The product reproduces the global three-dimensional atmospheric conditions from the ground to a 55 km height since 1979, with a temporal resolution of 6 hours and a spatial resolution of 30 km. The quality of the product is generally equivalent to that of international third-

generation global reanalysis products [39] (http://data.cma.cn/analysis/cra40, accessed on 5 November 2021).

ERA5 is the fifth generation ECMWF atmospheric reanalysis of the global climate covering the period from January 1950 to present. ERA5 combines vast amounts of historical observations into global estimates using advanced modeling and data assimilation systems. ERA5 provides hourly estimates of a large number of atmospheric, land, and oceanic climate variables. The data cover the Earth on a 30 km grid and resolve the atmosphere using 137 levels from the surface up to a height of 80 km [40]. Although ECMWF recently formed the global ERA5 dataset since 1950, the one from 1950 to 1978 is not the final version. There are some evaluation and application studies on the specific elements of the datasets at home and abroad which show that the quality of the datasets is significantly improved compared with the previous version [32,41,42]. Therefore, the hourly 2 m temperature data from ERA5 during 1979–2020 are used to form daily and monthly extreme air temperatures, annual frost days (FD), and hot days (HD) in mainland China. The ground spatial resolution of the reanalysis data is further improved to $0.25° \times 0.25°$.

*2.2. Methodology*

2.2.1. Seasonal Trend Analysis of Extreme Air Temperature

The seasonal trends of monthly maximum temperature (Txmax), monthly mean maximum temperature (Txmean), monthly minimum temperature (Tnmin), and monthly mean minimum temperature (Tnmean) were examined by seasonal trend analysis (STA) in mainland China. Seasonal trend analysis was initially applied to the trend analysis of image time series [43], and some scholars have used the methodology to analyze the minimum temperature over the La Plata River Basin in South America [44]. Due to the influence of solar radiation, atmospheric circulation, and other factors, meteorological elements change over time and can be considered a kind of fluctuation. For the time series of meteorological elements, we can regard it as the superposition of many harmonic waves [45]. A given time series $y_t$ and taking $t = 0$ as origin, $y_t$ can be decomposed into sine signals as Equation (1):

$$y_t = A_0 + \sum_{n=1}^{n=T/2} A_n \sin(\frac{2\pi nt}{T} + \phi_n) \tag{1}$$

in which $A_0$ is the arithmetic average value of the original series, and the other terms in the right hand side represent the $n$th harmonics over $T$. $A_n$ are amplitudes, and $\phi_n$ are phase angles (from 0 to 360°). $t$ is time, and $T$ is the temporal length of the series. The time series of any meteorological element is limited, and the maximum number of harmonics can be decomposed into half of the length of the series. Although a long-term series of meteorological elements contains a variety of time scale changes, the annual cycle is the most important; therefore, the first two harmonics can be used to simulate the original series [43]. Therefore, $n = 2$.

First, harmonic analysis of the temperature series was carried out, including five characteristic parameters, namely, the annual average amplitude (A0), the amplitude and phase of the annual cycle (A1 and F1), the amplitude and phase of the semi-annual cycle (A2 and F2) (Table 1) [21,43]. Here, A0 actually represents the annual average, A1 is the annual temperature range, F1 indicates shifts in time, where a value of 30° corresponds to 1 month approximately, and A2 and F2 are not clear, which can be regarded as the shape factor of the annual curve [44]. Second, once the five harmonic parameters were obtained for each year using Equation (1), their Theil–Sen median slopes were estimated. This slope was then used to characterize their trend. The significance of the statistics was evaluated using the non-parametric Mann–Kendall test ($p < 0.05$) [46]. Theil–Sen median slope estimation is a robust non-parametric statistical method that is insensitive to outliers and is very effective against reflecting the trend of time series data [47,48]. Finally, the trend of these parameters can be visualized. Since there are as many as $3^5$ combinations of these parameters, it is impossible to summarize all five seasonal curve shape parameters in

a single image. It is generally found that the three amplitude images contain the largest amount of information and that rendering trends in A0, A1, and A2 provide an effective composition. A companion phase trend is created by rendering trends in A0, F1, and F2. According to the classes of combination, the region of interest can be selected to draw the fitting curve of the beginning and ending years of climate elements, and the seasonal trend and change of elements can be better understood by combining with the image [30,48].

**Table 1.** Five characteristic parameters of harmonic analysis.

| ID | Name | Definition/Meaning |
|----|------|-------------------|
| A0 | annual average amplitude | the arithmetic average value of the original series/annual average temperature |
| A1 | amplitude of annual cycle | difference between maximum and minimum of the 1st harmonic/annual temperature range |
| A2 | amplitude of semi-annual cycle | difference between maximum and minimum of the 2nd harmonic/can be regarded as the shape factor of annual curve |
| F1 | phase of annual cycle | start phase angle of the 1st harmonic/indicate the time when the sine waves reaches a peak |
| F2 | phase of semi-annual cycle | start phase angle of the 2nd harmonic/can be regarded as the shape factor of annual curve |

The advantages of employing ERA5 and STA are (1) obtaining the spatial variation of extreme temperature, unlike the uneven station data; (2) obtaining five parameters of time series, that is, the change trend of temperature, the change of temperature range, and time change of maximum value can be understood at the same time; and (3) after the five parameters of the same grid point are superimposed and visualized, we can understand the temperature change pattern (different parameter combinations, different change patterns).

MATLAB and ArcMap were used for data processing and plotting, respectively.

### 2.2.2. Extreme Air Temperature Indices

We use five extreme temperature indices defined by the ETCCDI [31] and hot days (HD) that characterize extreme temperature [49,50]. Table 2 has listed the details of TN10p, TN90p, TX10p, TX90P, FD, and HD. The 90th and 10th percentiles of daily maximum/minimum temperature are calculated for a 5-day window centered on each calendar day in the base 1991–2020 period. In the last part of the paper, we evaluate the spatial distribution of the trend in extreme air temperature indices trends and examine the temporal evolution of the regional averages of these indices using linear trend rates. Linear estimation is a trend analysis method commonly used in climate analysis. In order to be able to compare with existing results and those of our article, this method was used here.

**Table 2.** List of extreme air temperature indices.

| ID | Name | Definition | Units |
|----|------|-----------|-------|
| HD | hot days | Annual count of days when TX $\geq$ 35 °C | days |
| FD | frost days | Annual count of days when TN < 0 °C | days |
| TN10p | cold nights | Percentage of days when TN < 10th percentile [1] | % |
| TN90p | warm night | Percentage of days when TN > 90th percentile | % |
| TX10p | cold days | Percentage of days when TX < 10th percentile [1] | % |
| TX90p | warm days | Percentage of days when TX > 90th percentile | % |

[1] TN/TX represent daily minimum air temperature/maximum air temperature.

## 3. Results

### 3.1. Seasonal Trends in Temperature

#### 3.1.1. Maximum Temperature

Figure 2 shows the spatial distribution of statistically significant trends in the annual average amplitude (A0), amplitudes of the annual (A1) and semi-annual cycles (A2), and phases of the annual (F1) and semi-annual cycles (F2) for monthly maximum temperature (Txmax) and monthly average maximum temperature (Txmean). Complementarily, a summary of areas with significant trends is presented in Figure 3.

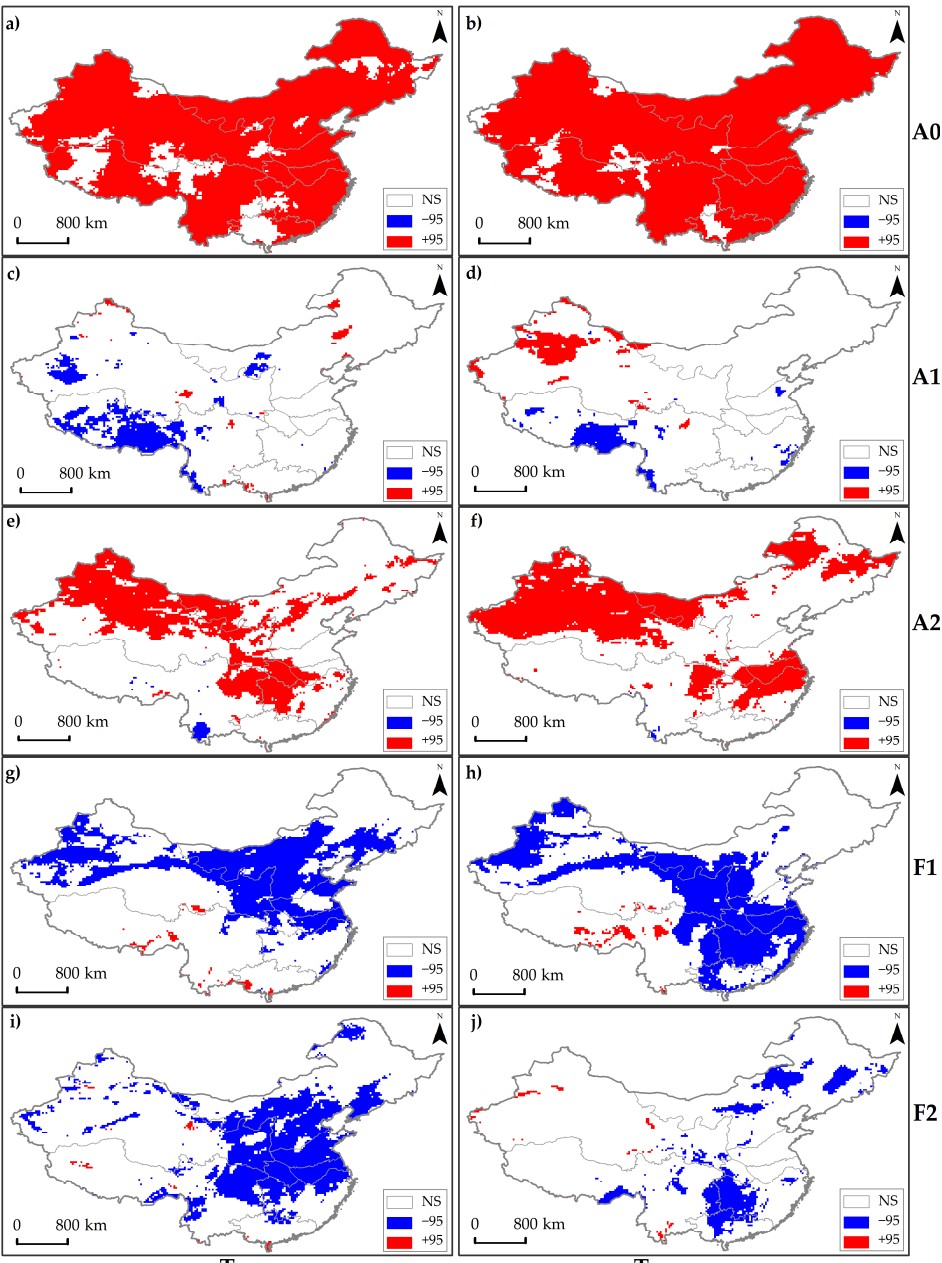

**Figure 2.** Areas in mainland China with trends of annual average amplitude (A0), the amplitude (A1) and phase (F1) of annual cycle, amplitude (A2) and phase (F2) of semi-annual cycle of Txmax (**a,c,e,g,i**) and Txmean (**b,d,f,h,j**). NS denotes no trend. −95 and +95 represent negative and positive trends, respectively, with a significant confidence level of 95% (same in Figure 7).

The amplitude variation in the maximum temperature was mainly positive, and the phase variation was negative over mainland China, as shown in Figure 2. During 1979–2020, the maximum temperatures in most parts of mainland China had a significant increase and there was no significant decrease. The A0 of Txmax and Txmean increased significantly in 84.5% and 93.2% of areas, respectively. The regions without significant change mainly occurred on the Tibetan Plateau and south China (Figures 2a,b and 3). Studies have revealed that the south is one of the regions with the weakest warming trend in China, and the warming trend on the Tibetan Plateau ranks first among the eight major regions in China [51]. In addition, the daily maximum temperature of the Tibetan Plateau from 1961 to 2015 had a warming trend [52]. The areas where Txmax and Txmean significantly changed in A1 accounted for 10.0% and 9.3% of the total area of mainland

China, respectively. The A1 of Txmax decreased significantly, mainly on the Tibetan Plateau and Tarim Basin. The area where the A1 of Txmean decreased significantly was slightly smaller than the area where it increased significantly. They appeared on the southeastern Tibetan Plateau, Tianshan Mountains, and Junggar Basin, respectively (Figures 2c,d and 3). Since the amplitude of the nonzero semi-annual is not easy to interpret, it may be related to the difference in the semi-annual period or annual curve shape in the seasonal curve [44]. Therefore, this paper only provides the results without analysis. Figure 2e,f show that the A2 of the maximum temperatures was approximately 1/3, showing a significant positive trend and mainly distributed in the northwestern region and Yangtze River Basin.

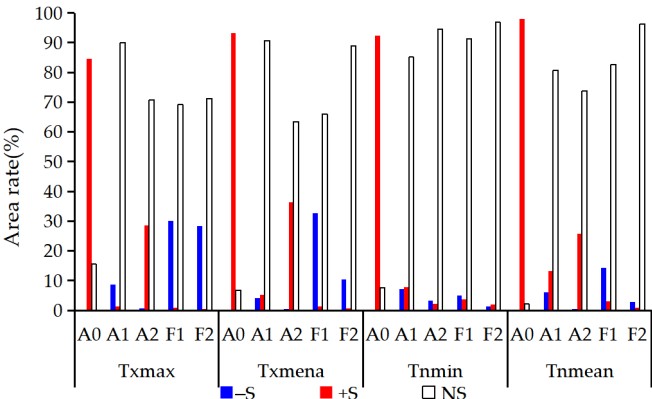

**Figure 3.** Area rate of mainland China with significant trends in the five parameters. +S/−S denotes positive/negative significant trends, respectively, with a significant confidence level of 95%.

F1 of the maximum temperature in mainland China, approximately 30% of the area, had a significant negative trend. Among these areas, Txmax mainly occurred in northwestern China and north of the Yangtze River, while Txmean had a pattern of 'shrinking in the north and expanding in the south', the area with a significant decrease in north China decreased, and the area with a significant decrease to the south of the Yangtze River increased (Figure 2g,h). F1 reflects the time when the sine waves reached a peak, which indicates that the time when the maximum temperature appeared in the above areas in the last 42 years was delayed. Some studies suggest that the phase change was related to a variety of mechanisms, but the influence of the change in thermal mass was greater [53]. Thermal mass on land is largely modulated by soil moisture. If soil moisture decreases, it will produce a positive phase shift [53]. Because of the lack of long-term and spatial high-resolution soil moisture datasets, it is very difficult to find conclusions supporting the above soil moisture and temperature changes from the existing studies on soil moisture changes in mainland China. The significant trend of F2 was also dominated by a negative trend, with 28.3% of Txmax decreasing significantly, mainly in the Yangtze River Basin and north China. The area where Txmean decreased significantly was approximately 1/3 of Txmax (Figure 2i,j).

Figure 4 shows the trends in monthly maximum temperature (Txmax) and monthly average maximum temperature (Txmean) during 1979–2020 (Figure 4a,b) together with the trends in A0 (Figure 4c,d). We can see that Txmax and Txmean in mainland China had a positive trend and the spatial distribution and magnitudes of the trend were very consistent with their A0. They had a significant linear relationship with a coefficient of determination of 0.92. This indicates that A0 from the seasonal trend analysis method, as a representative index of annual average temperature, is also suitable for the analysis of interannual temperature. From the perspective of spatial distribution, both Txmax and Txmean had a strong warming trend on the northeastern edge of the Tibetan Plateau, eastern coast, and Inner Mongolian Plateau. In addition, combined with Figures 2 and 4, the trend rates of the regions where the maximum temperature change was not significant were also small.

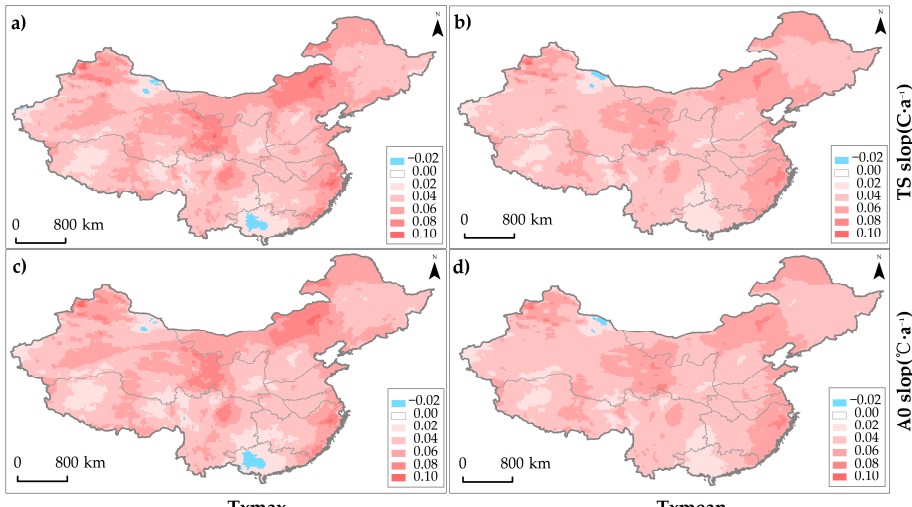

**Figure 4.** Theil–Sen trend (TS slope (**a**,**b**)) and linear trend for annual average amplitude (A0) of Txmax and Txmean (A0 slope (**c**,**d**)) during 1979–2020.

The five parameters of seasonal trends together represent the temporal dynamics of climate factors, up to 243 combinations. There may be one or several combinations with obvious advantages over mainland China. Therefore, the paper selects the first three significant combinations with the largest area from these combinations to examine the main classes and spatial distribution of the seasonal trend of each temperature element.

Figure 5 shows the first three classes of significant changes in Txmax and Txmean, which were characterized by a significant increase dominated by A0. The seasonal trend of Txmax was very distinct in mainland China, and only 9.6% of the areas did not change significantly. There were 63 significant change combinations and the first three accounted for 46.9% of the total area. A total of 28.7% (+0000, red) had a significant increase in A0 and no significant increase in other parameters. The combination is mainly distributed in the northeastern, southeastern coastal, and the Tibetan Plateau, indicating that while the annual average maximum temperature in these regions is increasing, the range of the annual maximum temperature and its occurrence time have not changed significantly.

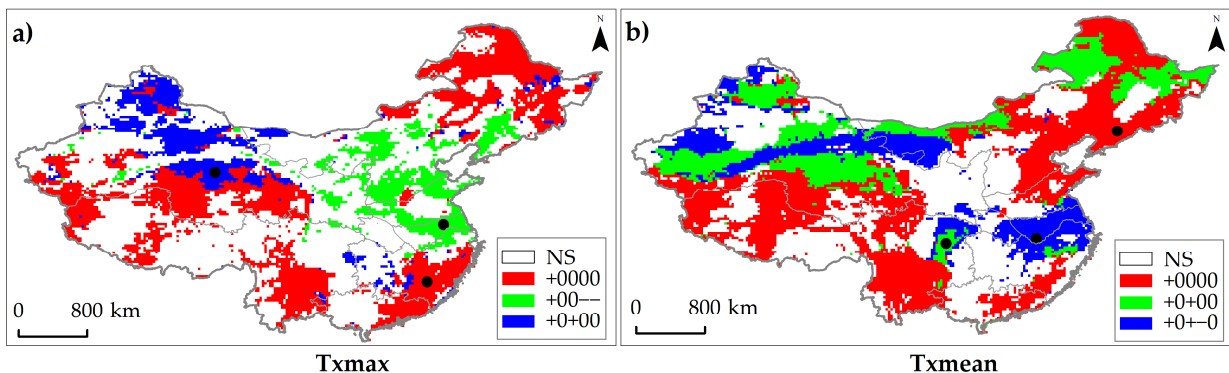

**Figure 5.** The combination of the top three areas of significant seasonal trends in Txmax and Txmean. Significant positive trends are marked with a "+" sign, significant negative trends with a "−" sign, and no-trends with a "0". Read from left to right, the symbols indicate the significance of trends in A0, A1, A2, F1, and F2. Black spots are randomly selected representative points of the three classes. For example, "+0000" has a significant positive trend in A0 and no significant trend in the other parameters.

The second combination accounted for 9.6%, with A0 increasing significantly, while F1 and F1 decreased significantly (+00−−, green), which was mainly distributed in the

eastern northwest China, northern Huang-Huai, and Jiang-Huai regions, indicating that the extreme maximum temperature in these regions generally increased and that the time was delayed. Both A0 and A2 also increased by 9.1% (+0+00, blue) and were mainly distributed in northwest China and the northern margin of the Tibetan Plateau (Figure 5a).

There were 51 combinations with significant seasonal variations in Txmean in mainland China, accounting for 96.3%. The first three classes were when A0 increased significantly (+0000), A0 and A2 increased significantly (+0+00), A0 and A2 increased significantly, and F1 decreased significantly (+0+−0), accounting for 34.8%, 16.2%, and 13.0% of the total area of mainland China, respectively. The first class was distributed mainly in northeast China, north China, the Tibetan Plateau, and the southeastern coast. The second was mainly in the northeast and northwest and the third appeared in the middle and lower reaches of the Yangtze River and northwest (Figure 5b). It can be concluded that in some regions of Huang-Huai and Jiang-Huai, the maximum temperature not only had a significant upward trend but also the time at which its maximum value appeared was significantly delayed. These two places are one of the main grain-producing regions in China, which provides some ideas for follow-up studies on the effect of temperature increases on grain yield.

A grid was randomly selected from the first three classes of Txmax and Txmean, and the monthly dynamics of the start year (1979, black curve) and end year (2020, red curve) were fitted (Figure 6). We can see that the shapes of the curves are different mainly due to different parameter combinations. Even if the same class is different due to locations and elements, the seasonal trend of the same class of curves is consistent, regardless of their shapes. For example, when A0 increased significantly (+0000), the overall value in 2020 was higher than that in 1979. When F1 decreased significantly, the peak time was obviously delayed (+00−−, +0+−0). However, the curve of a significant increase only in A0 of Txmax seems to have significantly delayed in 2020, but the statistical test is not significant, which should be related to the large difference in the time of the maximum at this grid.

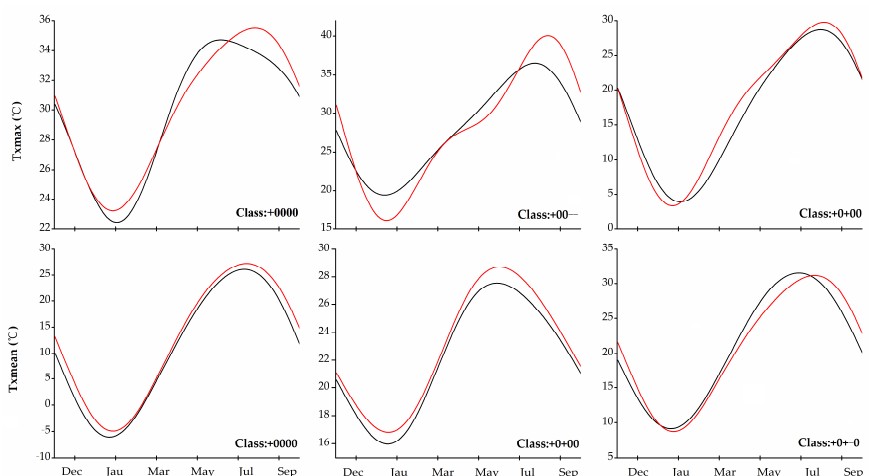

**Figure 6.** Examples of seasonal trends representative of each of the three major trend classes in Txmax and Txmean. The black curve represents the seasonal curve at the beginning of the series while the red curve represents the seasonal curve at the end of the series. The difference between the two indicates the change resulting from the trend.

### 3.1.2. Minimum Temperature

Similar to the maximum temperature, the minimum temperature also had a significant increase in mainland China (Figure 7). The A0 of monthly minimum temperature (Tnmin) and monthly average minimum temperature (Tnmean) increased significantly in 92.4% and 97.9% of areas, respectively. From this point of view, the warming of the minimum temperature was larger than that of the maximum temperature, which is consistent with existing studies [54,55], but the time variation of its minimum value was slightly smaller than that of the maximum temperature (Figure 7g,h). The regions where A0 of the minimum

temperature did not change significantly were scattered on the Tibetan Plateau, northwest China, and northeast China (Figure 7a,b). In contrast, the minimum temperature in the mid-lower reaches of the Yellow River, Yangtze River Basin, Jiang-Nan, south, and eastern southwest China showed a positive trend in the last 42 years. The above regions are major agricultural areas in China. The regions where A1 of Tnmin and Tnmean increased and decreased significantly were bounded by the 400 mm isohyet in mainland China, i.e., the temperate continental and plateau mountain climatic areas mainly increased, while the monsoon climatic areas mainly decreased (Figure 4c,d). Comparing the spatial distribution of the parameters of maximum temperature and minimum temperature, we can find that A0, A1, A2, and F1 of maximum temperature, A0, A1, F1, and F2 of minimum temperature, and their trend of Txmax/Tnmin and Txmean/Tnmean had similar spatial patterns. However, there was an exception. A2 of Tnmin showed no change in most regions, while Tnmean showed a significant increasing trend in northern China (Figure 7e,f).

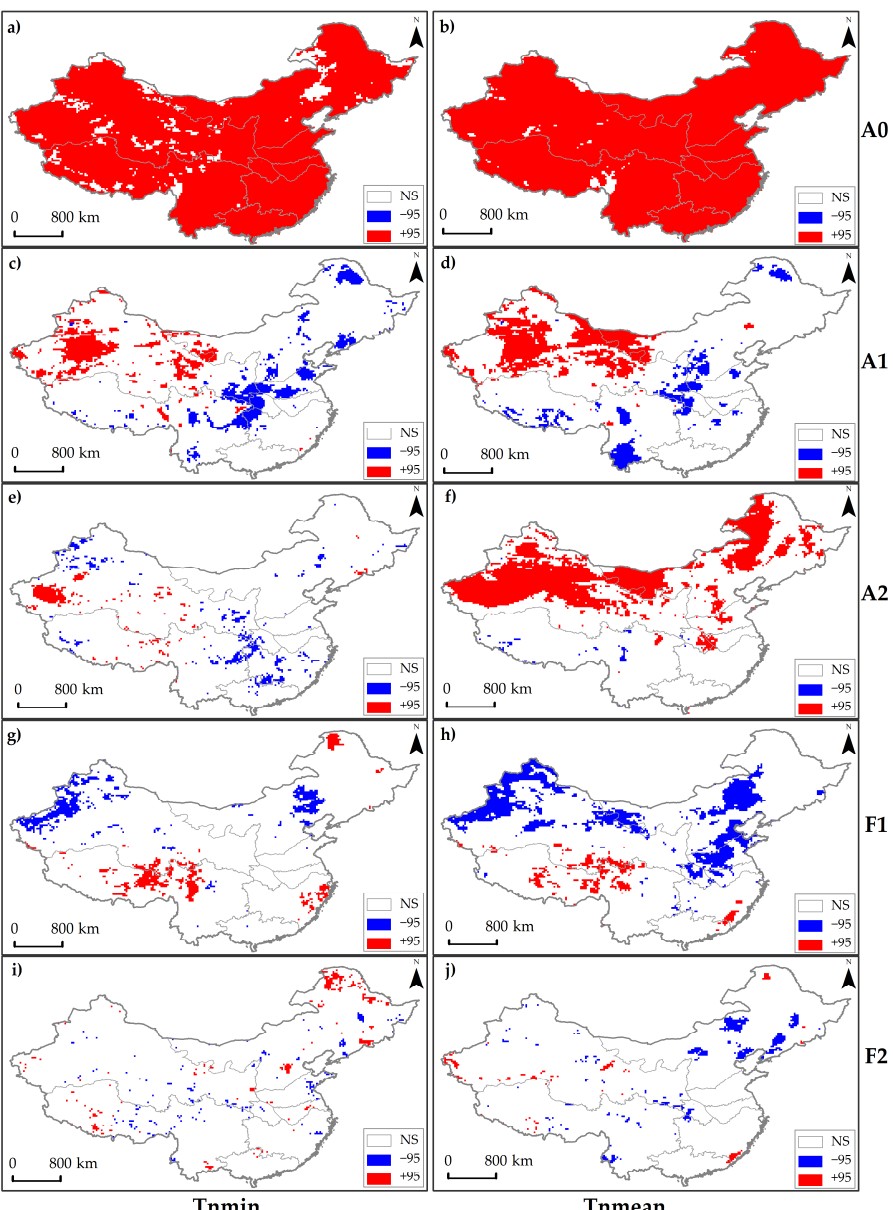

**Figure 7.** Areas in mainland China with trends of annual average amplitude (A0), the amplitude (A1) and phase (F1) of annual cycle, amplitude (A2) and phase (F2) of semi-annual cycle of Tnmin (**a,c,e,g,i**) and Tnmean (**b,d,f,h,j**).

F1 of Tnmin had a significant change in 8.6% of the area and the area with a significant increase was slightly smaller than that with a significant decrease (Figure 7g). F1 of Tnmean decreased significantly in 14.4% of the area and increased significantly in 3.1% of the area (Figure 7h). The area where F2 changed significantly was further reduced, accounting for 3.2% and 3.7%, respectively (Figure 7i,j).

In the last 42 years, the spatial distribution and magnitude of the trend of minimum temperature were also similar to the trend of their A0, and linear regression determination coefficients were 0.96 and 0.94, respectively (Figure 8a–d). The warming trends of Tnmin and Tnmean were generally higher in the north and lower in the south. The warming trend rate for most parts of the north was 0.04–0.08 °C·a$^{-1}$, and that for the south was not more than 0.04 °C·a$^{-1}$. The spatial patterns of the warming trends of Tnmin and Tnmean were also similar, and the warming trend rate of the former was higher than that of the latter (Figure 8).

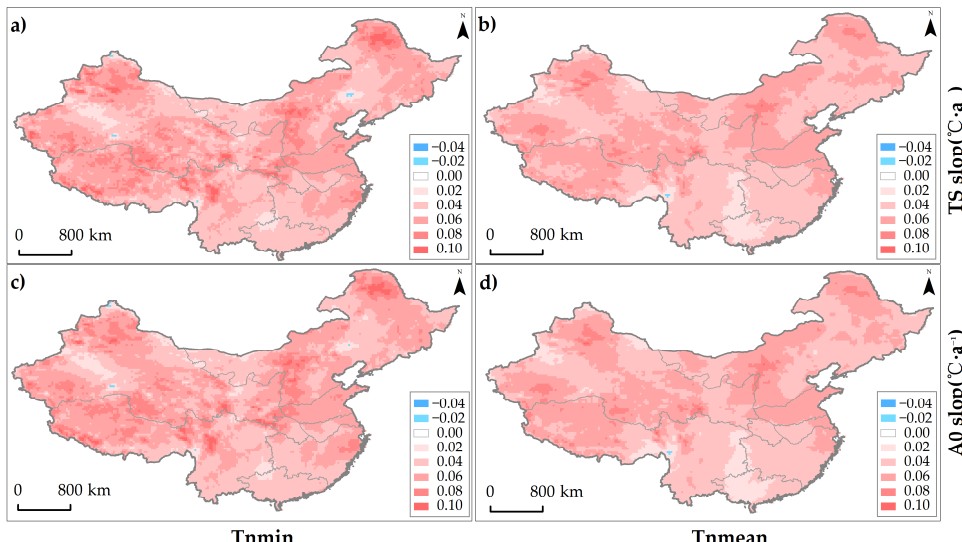

**Figure 8.** Theil–Sen trend (TS slope (**a**,**b**)) and linear trend for annual average amplitude (A0) of Tnmin and Tnmean (A0 slope (**c**,**d**)).

The seasonal trend of Tnmin was very distinct in mainland China and all were mainly characterized by significant changes in amplitude, which was somewhat different from the maximum temperature. A total of 94.7% of the areas had significant changes, including 57 combinations. The first three classes of significant change accounted for 78.1% of mainland China, and the first class (+0000) was the most distinct, accounting for 67.6% of the total area. The A0 and A1 classes increased (++000) and the A0 increase and A1 decrease (+−000) accounted for 5.3% and 5.1%, respectively, and they appeared in the western and central regions, respectively (Figures 9a and 10).

The first three classes with significant seasonal trends in Tnmean were also dominated by amplitude, and all had increased significantly. A0 increased significantly (+0000), and both A0 and A2 increased significantly (+0+00). The three amplitudes all increased significantly (+++00), accounting for 52.2%, 12.6%, and 5.8% of mainland China, respectively, and the latter two mainly appeared in northwestern China (Figures 9b and 10).

From the three fitting curves, we can see the seasonal trends of Tnmin and Tnmean. Because these three classes are amplitude combinations, the phase change was no-trend at the beginning year and the end year; that is, the time of peak appearance was no different (Figure 10).

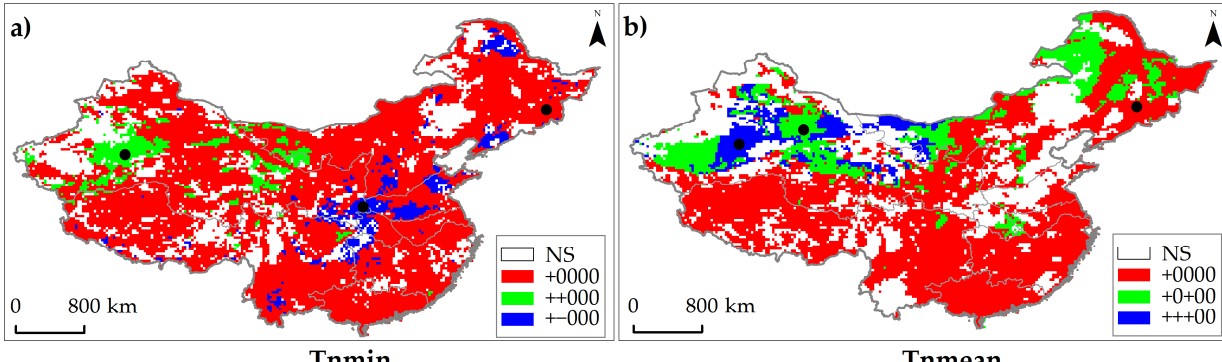

**Figure 9.** The combination of the top three areas of significant seasonal trends in Tnmin and Tnmean. The meaning of these symbols is the same as that in Figure 5.

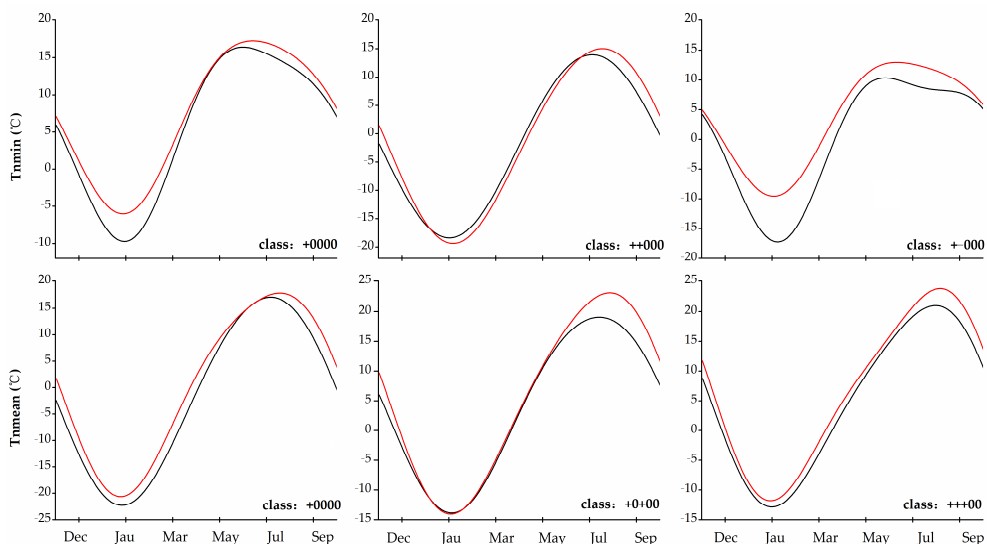

**Figure 10.** Examples of seasonal trends representative of each of the three major trend classes in Tnmin and Tnmean.

*3.2. Change of Extreme Air Temperature Days*

3.2.1. Hot Days, Cold Days, and Warm Days

During 1979–2020, hot days (HD) increased significantly in the eastern and northwestern regions (22.7%), with trend rates of 0.2~0.6 d·a$^{-1}$ and 0.6~0.8 d·a$^{-1}$ in some regions of the lower reaches of the Yangtze River. The area of significant decrease was small (5.3%), mainly in the northeast (Inner Mongolia and parts of Liaoning), and their trend rate was not less than $-0.4$ d·a$^{-1}$ (Figure 11a,b). HD in most parts of the north, northwest and Tibetan Plateau had no significant change, which is related to the fact that there were few or no temperatures higher than 35 °C.

TX10p decreased significantly in eastern Chain, eastern northwest China, and most of the Tibetan Plateau. Most other regions had a decreasing but not significant trend. The trend rate of the significant reduction was $-0.4$–$-0.2$%d·a$^{-1}$. In western Xinjiang, there was an increasing trend and a significant increase in the Tianshan region, but the trend rate did not exceed 0.2% d·a$^{-1}$ (Figure 11c,d). TX90p had a significant increase in most regions, and the trend was not significant in adjacent areas of the south, southwest and Jiang-Nan, eastern northwest, Tibetan Plateau, and most of the Tarim Basin. In terms of the spatial distribution of the increasing trend rate, most regions were less than 0.4% d·a$^{-1}$, and the increasing trend was slightly prominent in the eastern northwestern and southern southwestern regions (Figure 11e,f).

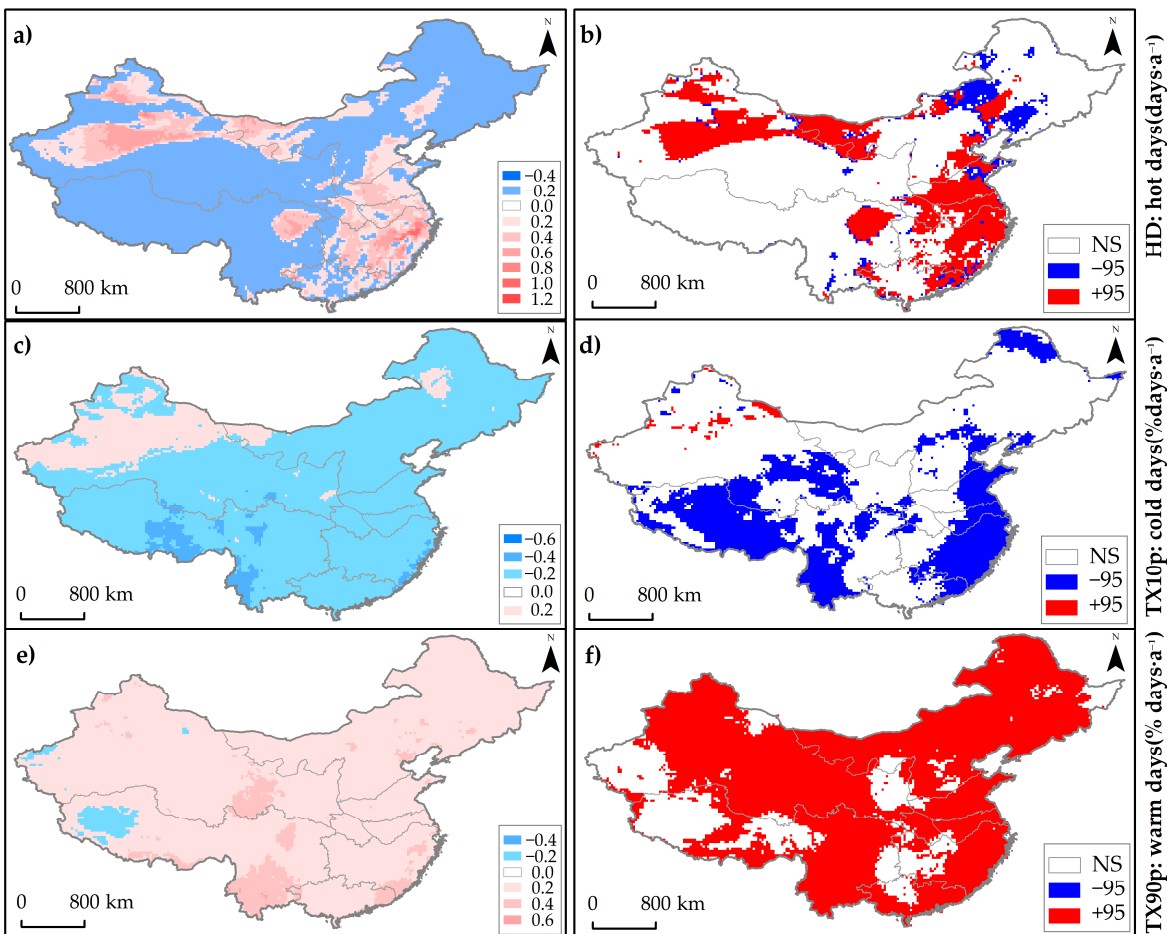

**Figure 11.** HD (**a**,**b**), TX10p (**c**,**d**), and TX90p (**e**,**f**). Left for trend rate and right for statistically significance.

The regional average of HD, TX10p, and TX90p show that HD and TX90p had an obvious increasing trend, while TX10p had a decreasing trend (Figure S1). The average HD does not seem to have been high in mainland China. This is mainly due to the vast territory of China, with large differences from east to south and from north to south. Some regions in the northeast and Tibetan Plateau do not experience daily temperatures above 35 °C, while the southeast and Turpan Basin may have temperatures as high as 40 days, which further indicates that HDs in warm regions have increased significantly. From the regional trend rate, HDs and TX10p were not as large as TX90p, which was mainly related to the increase in TX90p in most of mainland China.

### 3.2.2. Frost Days, Cold Nights, and Warm Nights

Frost days (FD) mainly occur in winter, early spring, and late autumn in China. HD is closely related to latitude and altitude. For example, some regions on the Tibetan Plateau have frost year-round, while most regions in south China have frost-free days for approximately 350 days out of the year. During 1979–2020, the change in FD was not significant in most of the area south of the Yangtze River, and the decreasing trend north of the Yangtze River was significant. In addition to the change in FD, this distribution pattern may also have been related to a few HD in the south. In the regions where FD decreased significantly, the trend rate was mostly −0.2–0.8 d·a$^{-1}$, and it could reach −0.8–1.0 d·a$^{-1}$ in some regions on the Loess Plateau and Tibetan Plateau (Figure 12a,b).

TN10p decreased significantly in most regions, but they were not significant in the northwestern and northeastern regions. In western Inner Mongolia and the Tarim Basin, there were no trends. The trend rate of a significant decrease in most regions did not exceed 0.2% d·a$^{-1}$ (Figure 12c,d). Warm nights (TN90p) had a significant increasing trend in most

regions, but the increase was not significant only in some regions of southern Jiang-Nan and western Jiang-Han and east of southwest China. In terms of the spatial distribution of trend rates, TN90p increased more significantly in western than in eastern China (Figure 12e,f).

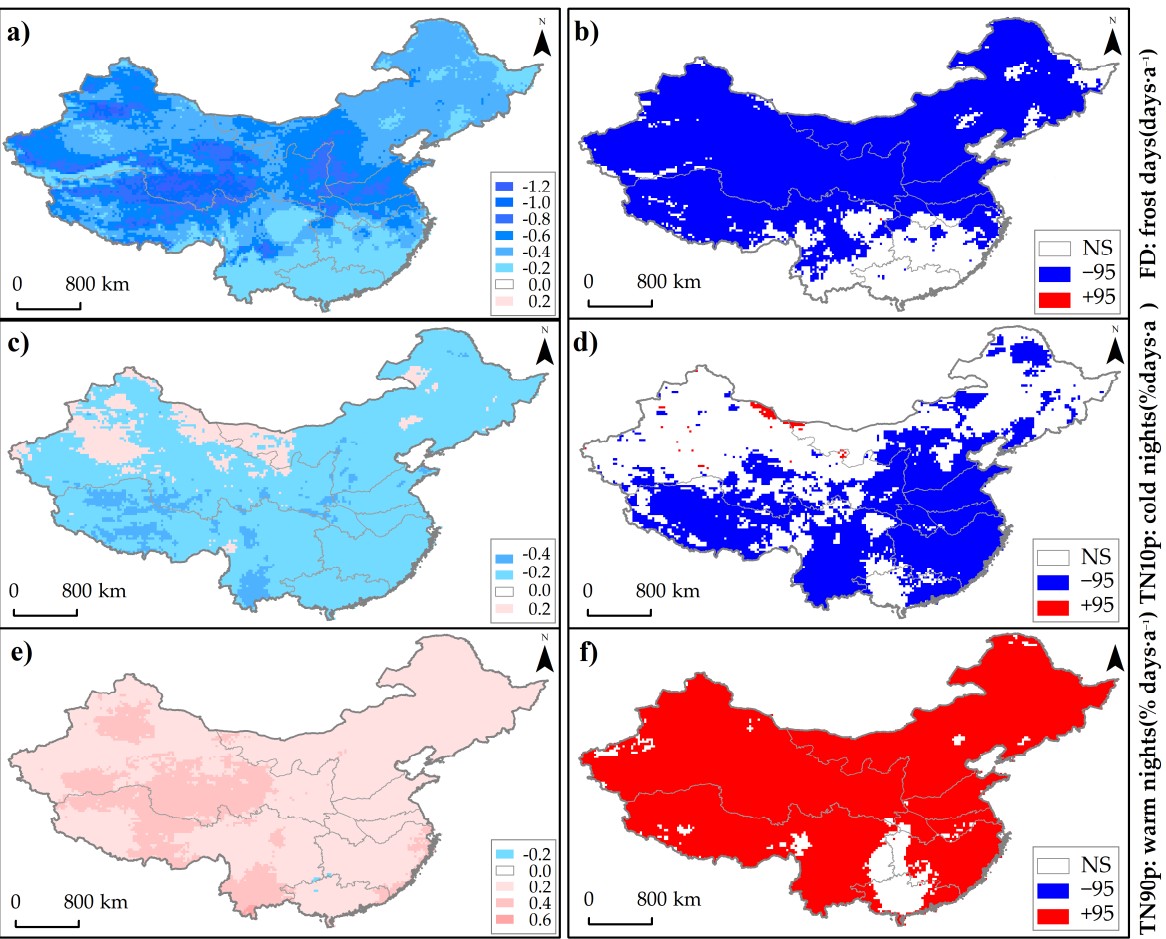

**Figure 12.** FD (**a**,**b**), TN10p (**c**,**d**), and TN90p (**e**,**f**). Left indicates trend rate and right for significance.

After the regional average in mainland China, both FDs and TN10p had a decreasing trend, and TN90p had an increasing trend. From the absolute value of the trend rate, TN10p was smaller than TN90p, which shows that the warmer minimum temperature increased more distinctly (Figure S2).

The present studies on extreme temperature changes in mainland China show that although the most extreme high temperatures were increasing and extreme low temperatures were decreasing, there were certain differences between regions and magnitudes. It has been reported that days of extreme temperatures at some observatories in mainland China do not conform to a normal distribution [56–59]. Therefore, this difference may be related to the methods and data.

## 4. Discussion

The reanalysis dataset can provide grid data with uniform spatial distribution and its surface temperature has high reliability [39,60,61]. The extreme temperatures are better captured by the ERA5 [60]. Based on the ERA5, we analyzed the seasonal trend of monthly extreme maximum and minimum temperature in mainland China during 1979–2020 and the time evolution of extreme air temperature days in this paper, hoping to provide help for studying the long-term impact of climate change on grain production.

Under the indisputable fact of climate warming, whether it is a single observation site, reanalyzed data, or model predictions, it was found that extreme warm events have increased significantly, while extreme cold events have decreased significantly in China [1,62–67]. This trend is consistent with the changing characteristics of global temperature extremes [62]. In recent years, some studies have attempted to interpret and assess the effects of extreme climate events on crop yield [19,68–70]. Due to the lack of long-term trials or yield observation data, China's research on this aspect is more based on crop simulation models or statistical yield, or short-term experiments [17,19,62,63,68,70]. The results show that an increased heat ETS (extreme temperature stress) and a decreased cold ETS would be expected for most areas of China during 2020–2049, and the spatial variability of rice yield loss will be greater [63]. Single rice in northeast China and early rice in south China will be under severe cold stress, while single rice in the middle and lower reaches of the Yangtze River and late rice in south China will be under severe heat stress [63]. During 1986–2015, the multi-year average extreme high temperature days increasing every additional day could result in a 226.62 kg·hm$^{-2}$ multi-year average maize yield reducing in the main summer maize cultivating area of China [68]. The number of extreme high temperature days has shown an increasing trend during 2021–2050 under RCP4.5 and RCP8.5, which could result in maize yield decreased by 9.2% and 27.3%, respectively [68]. Based on analysis of more than 20,000 historical maize trials, Lobell et al. concluded that each degree day spent above 30 °C reduced the yield by 1% under optimal rain-fed conditions and by 1.7% under drought conditions [24,69]. The conclusion proves the effect of extreme high temperature with experimental data.

A study has pointed out that the yield of maize and rice will decrease with the increase of temperature in the mean growing season [70]. Conversely, the maize and rice yield would increase by approximately 6.947% and 2.885% with a 1 MJ·m$^{-2}$ increase in the mean growing season downward shortwave solar radiation, respectively [70]. Furthermore, radiation is greatly affected by cloud cover, which is very important for agricultural production, crop distribution, and animal migration. With climate change, global cloudiness has also changed significantly [71–74]. Limited by space, this article only analyzed the seasonal trend of extreme temperature in mainland China and did not assess the impact of the trend on the crop yield, especially rice, maize, and wheat. Past studies have focused mainly on the number of days of extreme temperature changes and have paid less attention to changes in the monthly or annual range of extreme temperatures, as well as changes in the time of occurrence [49,65,67,75–78]. For these reasons, the result has certain significance for future studies on the impact of extreme climate on agricultural production. In addition, the effects of extreme temperatures, moisture conditions, and cloud cover on crop yields are comprehensive, and future research should focus on their compound and dynamic effects.

## 5. Conclusions

Monthly maximum temperature (Txmax) and monthly mean maximum temperature (Txmean) had the same annual cycle change with a significant positive trend. The five characteristic parameters for annual and semi-annual cycles in Txmax and Txmean were similar in spatial distribution, indicating that Txmax and Txmean in most parts of mainland China had analogously seasonal variations. The trend in annual average amplitude (A0) was largest, and the annual amplitude (A1) was smallest. The areas with significant changes in annual phase (F1) and semi-annual phase (F2) mainly decreased, which indicates that the time of maximum temperature in these regions had a delaying trend. The area with a higher trend rate in Txmax was larger than that in Txmean, but the trend rate of both was less than 0.06 °C·a$^{-1}$ in most regions, showing a strong warming trend on the northeastern edge of the Tibetan Plateau, eastern coast, and Inner Mongolian Plateau. The maximum temperature changed significantly to over 90% of mainland China and the changes in A0 were dominant.

The A0 values of monthly minimum temperature (Tnmin) and monthly mean minimum temperature (Tnmean) in most of mainland China also had a significant warming

trend; the trend rate was 0.02–0.08 °C·a$^{-1}$, which was higher in the north than in the south, and the Tibetan Plateau was especially prominent. The Tnmin warming trend was higher than that of Tnmean. The significant change area of A1 was significantly smaller than that of A0, dispersing on both sides of the 400 mm isohyet; that is, the northwestern area mainly increased and the southeastern area mainly decreased. Different from the maximum temperature, the area where F1 with the minimum temperature changes significantly decreased and the areas with increasing trends increased. The change in A0 in minimum temperature was also dominant and its proportion was higher than maximum temperature, which shows that the trend in minimum temperature in mainland China was more distinct than maximum temperature.

In recent years, the number of heat waves has increased [75–78]. HD has increased significantly in the eastern and northwestern regions (significantly increased areas account for 22.7% of mainland China). However, there was no significant change in those areas where HD may have occurred in south China, east of southwest China, south of north China, and northwest China. TX90p had a significant increase in most regions, while TX10p had a significant decrease on the eastern Tibetan Plateau, most of the Tibetan Plateau, and eastern northwest China. In particular, FD decreased significantly on the Tibetan Plateau. TN10p decreased significantly in most regions but did not change significantly in the northwest and northeast. TN90p increased significantly in most regions.

**Supplementary Materials:** The following are available online at https://www.mdpi.com/article/10.3390/su132212462/s1, Figure S1: Temporal evolution of regional averaged HD, TX90p, and TX10p and Figure S2: Temporal evolution of regional averaged FD, TN90p, and TN10p.

**Author Contributions:** J.Z. and J.L. contributed to the study conception and design. Material preparation, data collection, and analysis were performed by W.Y. and Y.W. The first draft of the manuscript was written by W.Y. and all authors commented on subsequent versions of the manuscript. All authors have read and agreed to the published version of the manuscript.

**Funding:** This research was funded by Foundation Key Research and Development Program of Ningxia Hui Autonomous Region, 2020BBF03009; Key Research and Development Program of Ningxia Hui Autonomous Region, 2020BBF03024; and Natural Science Foundation of Ningxia Hui Autonomous Region, 2020AAC03467.

**Institutional Review Board Statement:** Not applicable.

**Informed Consent Statement:** Not applicable.

**Data Availability Statement:** ERA5 daily temperature data used in this paper are available at ECMWF website (https://www.ecmwf.int/en/forecasts/datasets/reanalysis-datasets/era5, accessed on 4 November 2021).

**Acknowledgments:** We thank the reviewers for their help in improving this manuscript.

**Conflicts of Interest:** Authors declare no conflict of interest.

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
