# Peer review of "Assessment of Seasonal Variability of Extreme Temperature in Mainland China under Climate Change"

_sustainability, doi:10.3390/su132212462_

Round 1

Reviewer 1 Report

Dear Authors,

My comments aim to improve the quality of this manuscript. Please receive my comments in a welcoming and constructive manner.

The manuscript aims to evaluate the distribution of maximum and minimum air temperatures and hot days, frost days, warm days, cold days, warm nights and cold nights using seasonal trend analysis in Mainland China in 1979-2020.

Lines 20-21: Here and elsewhere in the text. Use different ID rather than A1, F1, A0; these ID are confusing and do not facilitate the reading of the manuscript. Use IDs connected to the definitions/indices.

Line 20: What is phase F1? See the previous comment.

Lines 21-22: Is this trend statistically significant?

Line: 23-24: Provide the values of annual amplitudes for minimum and maximum air temperatures.

Line 25: “400mm isohyet”. Not clear as the first part of the sentence referrer to maximum and minimum air temperatures. Clarify this sentence.

Lines 26-27: Are the increases in HD, Tx90p and Tn90, and the decreases in FD, Tx10p and Tn10p statistically significant? Mention the regions.

Lines 27-28: This sentence is not clear; please re-write it. How much has mean air temperature increased over mainland China in the past 42 years? Is this trend statistically significant, and at what level of significance?

Lines 31-101: Introduction section. What are the projections for maximum temperature, minimum temperature, frost days, hot days, warm nights, cold nights, warm days and cold days for China? A few sentences supported by references should be included since these indicators are assessed in the manuscript.

Lines 32-33: Cite the new IPCC report, which was released this year.

Line 38-39: This information is not presented on the map. Add an inset map with the location of China.

Lines 48-49: Reference required. Indicate how much has the temperature risen in the last 50 years in China. Is the air mean temperature trend statistically significant in China, and what is the level of significance?

Lines 50-53: Reference required.

Lines 50-54: Does the trend in the growing season length in China follow the global trend? Indicate it in the text by adding a sentence and a reference. Is this trend statistically significant in China, and what is the level of significance?

Lines 55-57: Does the trend in the hottest day and coldest night in China follow the trend for the globe? Indicate it in the text by adding a sentence and a reference. Are the trends statistically significant in China, and what is the level of significance?

Lines 57-61: Indicate the trend for cold days, cold nights, warm nights and warm days. Are these trends statistically significant, and at what level of significance? Do the trends in cold days, cold nights, warm days and warm nights in China follow the trends for the globe? Indicate it in the text by adding a sentence and a reference.

Line 63: Remove “An increase in” as the authors cited a decreasing trend in the cold days and cold nights in the previous lines.

Lines 66-67: Not clear; please re-write the sentence. Reference required.

Lines 67-68: Reference required. What places? Provide examples and respective references.

Lines 74-75: References required.

Lines 76-78: References required.

Line 86: No clear, please re-write the sentence.

Line 88: “aroused” – chose a different word.

Lines 89-90: Provide examples of previous studies with references.

Lines 90-92: Provide examples of previous studies with references.

Lines 93-95: Repetitive of the content in lines 55-65. Re-write the sentence and add references.

Lines 95-97: Mention what indices are you studying in this manuscript. Add a reference or online link, so the readers can access the full list of the ETCCDI indices.

Lines 100-101: The sentence is not clear; re-write it.

Line 101: A hypsometric map would be better to explain the differences in the maximum and minimum air temperature and extreme air temperature indices in the results section.

Lines 117-123: Provide references for these reanalysis datasets. Briefly characterise the ERA5 main reanalysis.

Lines 124-128: Provide a reference.

Lines 132-134: Not clear; please re-write the sentence.

Line 135: ERA5.

Lines 141-172: Why use this methodology and not a different one? The methodology section requires clarification.

Lines 146-149: Not clear; please re-write the sentence.

Lines 154-161: What is annual and semi-annual periodicity? What is annual average amplitude, annual amplitude, annual phase, semi-annual amplitude and semi-annual phase? The definitions are not presented. The IDs A0, A1, F1, A2 and F2 are not clear and do not facilitate the reading of the manuscript.

Lines 154-172: What software was used to calculate the analysis? Please indicate it.

Lines 161-162: Provide a reference for the Mann-Kendall test and possibly the formula. Indicate the level of statistical significance. What software was used to calculate the Mann-Kendall test?

Lines 162-164: Why using the Theil-Sen median slope? Why using this methodology and not a different one?

Line 173: Remove “number” and Re-write the title for this subsection. For example, “Extreme temperature” or “Extreme air temperature indices”.

Lines 174-182: The definitions of these indices would better be presented in a table with columns such as: ID – FD, Name – Frost days, Definition, Units – days.

Lines 176-178: Provide a reference for the definition of hot days. Is this a definition by the Chinese Meteorological Administration or other references?

Lines 179-180: What is the standard climate period (climate normal) used in this research? Please indicate it.

Lines 174-186: What software was used to calculate the analysis? Please indicate it.

Line 189: Section 3.3.1. Maximum temperature – it would be interesting to analyse per season (spring, summer, autumn, winter).

Lines 203-204: Mention the value of the warming trend. Is this trend statistically significant, and at what level?

Lines 213-216: Not clear.

Lines 227-228: “Some people think”. Not scientific language! Re-write the sentence.

Line 229: Add references.

Lines 238-239: These questions should be removed from the text and the answers inserted.

Line 241: Add the orientation and the scale to the maps.

Line 258: Add the orientation and the scale to the maps.

Lines 259-263: The paragraph is not clear. The questions should be removed and the answers inserted.

Lines 271-274: Not clear.

Line 305: Add the orientation and the scale to the maps.

Lines 305-309: The three cases are not clear. Re-write the classes.

Line 311: The graphs & classes are not clear.

Line 312: Section 3.3.2. Minimum temperature – it would be interesting to analyse per season (spring, summer, autumn, winter).

Line 341: Add the orientation and the scale to the maps.

Line 351: Add the orientation and the scale to the maps.

Line 371: The three cases are not clear. Re-write the classes.

Line 373: The graphs & classes are not clear.

Line 375: Section 3.2.1 Hot days, cold days and warm days – it would be interesting to analyse per season.

Line 403: Add the orientation and the scale to the maps.

Line 404: Section 3.2.2. Frost days, cold nights and warm nights – it would be interesting to analyse per season (spring, summer, autumn, winter).

Line 433: Add the orientation and the scale to the maps.

Lines 434-474: Do these findings follow state of the art for the globe? What are the implications of these findings for the agriculture of different regions in China? The manuscript stated in the introduction how climate change affects agriculture; however, the paper lacks a discussion of the findings and their link with agriculture. The paper focuses on the impacts on agriculture only. However, increasing trends in warm nights and hot days impact health and mortality, for instance.

Line 466: References required.

Overall comment: Research on trends of maximum and minimum air temperatures and ETCCDI indices in Mainland China is not new since many research papers have been published on these subjects.

Reviewer 2 Report

In summary, I think that this manuscript could be accepted as an article in the Journal Sustainability after revision. This article has a big importance for readership. 

Good luck to the authors 

The Reviewer#3

Round 2

Reviewer 1 Report

Dear Authors,

Thank you for accepting my previous revisions. The quality of this manuscript has improved. However, additional revisions are required to achieve higher scientific standards as required by the journal Sustainability.

Please kindly consider my revisions below:

Lines 20-21: Unclear.

Lines 24-25 and 26: Replace “number of days” with “extreme air temperature indices”. HD and FD are duration indices expressed by the number of days above and below a specific threshold, respectively, but TX90p, TN90, TX10p and TN10p are percentile indices expressed as percentage days according to the ETCCDI.

Lines 31-32: Provide the reference at the end of the sentence.

Lines 33-35: Correct the citation. Additionally, the citations in lines 98, 120, 132, 139 and 184, which have an online link, might have to be changed to a number.

Lines 50-51: Remove “total amount”. Provide reference.

Line 51: Replace “surface temperature” with “surface air temperature”.

Lines 57-58: Provide reference.

Line 65: “extreme temperature”. Clarify the air temperature indices by adding the name to the sentence.

Line 60: Remove Latvia to focus on southwest China.

Lines 71 and 74: “high temperatures”. Are you referring to the mean or maximum air temperature? Please clarify it in the text.

Line 98: The link is not working.

Line 132: Check the link.

Lines 148-181: Seasonal trend analysis of extreme air temperature. Require more explanation on the methodology and formulas so that other researchers can replicate this methodology.

Lines 246-247, 263, 346, 356: Add the names of the A0, A1, A2, F1, F2 in the captions under the maps for better reading. E.g. mean annual amplitude (A0), etc.

Lines 154 – 156: “Temperature is affected by solar radiation, atmospheric circulation and other factors”. It requires a reference. Also, “other factors” is not clear.

The first part of the sentence does not fit the last part “can be regarded as the superposition of many harmonics in time”. Please clarify/re-write it.

Lines 156-160: Add references.

Line 161: Add references to the harmonic analysis.

Lines 162-166: The definitions of A0, A1, F1, A2, F2 must be clearly stated and perhaps should be indicated on a table for clarity. The table could include the following columns: ID – e.g. A0, Name – e.g. Annual average amplitude, Definition, Reference.

Lines 183-196: Table 1 is not cited in the text.

Lines 185-189: This information is presented in the table.

Line 197: Table 1 – Replace “daily maximum temperature” with TX for consistency.

Line 198: daily minimum air temperature/maximum air temperature.

Line 235: Remove the word “think” as it is not adequate in a scientific context. Replace this word for “suggest”, for instance.

Lines 235-237: References are required.

Lines 263, 356: Is this the trend per year? Clarify in the maps and under the caption. Perhaps the Thei-Sen trend and the linear trend should be provided for the period 1979-2020, as the trends would be more explicit. The trends in the text would have to be changed too.

Why use the linear trend? This was not explained in the methodology; only the Theil Sen trend is mentioned in the methodology. Clarify.

Lines 310, 316, 376, 378: As per previous revisions, the classes +0000, +00--, +0+00 are extremely confusing and do not facilitate the reading of the text. The authors should choose different classes to represent positive, negative and no trends, and statistical significance. In the scientific literature, 0 would refer to non-trend instead of “insignificant trend” as stated in the text in line 311. The classes need to be changed in the text as well.

Lines 446-448: This sentence is too general. This study employed ERA5 data; however, instrumental data and model predictions are mentioned. References on previous studies are required in this sentence.

Lines 450-451: “some studies”. References are required.

Lines 452-453: Provide references/examples of such studies.

Lines 454: Clarify heat extreme temperature stress and cold extreme temperature stress. How are they related to the extreme air temperature indices analysed in this manuscript?

Lines 458-461: References required.

Line 464: Change reference to number.

Line 468: “Some studies”. Provide references.

Lines 474-476: Clarify. Would you suggest including it in future research? Would this research be conducted by the authors of the present study or other authors? Are there any databases on crop yield such as rice, maize and wheat productions in China?

Lines 476-479: “Past studies”. Provide references.

Line 488: “homogenous”. A different word would be appropriate for this context.

Overall comment: As per my previous revision, research on trends of maximum and minimum air temperatures and ETCCDI indices in Mainland China is not new since many research papers have been published on these subjects. The authors have replied to this. However, this should be clearly stated in the manuscript. What are the advantages of employing reanalysis data and seasonal trend analysis? What are the contributions of this study in comparison to previously published research articles? Do the results of this research follow the results of previously published papers although using different methodologies?

Author Response

"Please see the attachment”

Round 3

Reviewer 1 Report

Dear Authors,

Thanks for accepting my previous suggestions. The manuscript has now better quality in comparison to previous versions.

A few comments are below:

Line 188 – table 1: Check the English.

Line 269 (figure 4), 365 (figure 8), 418 (figure 11), 448 (figure 12): Confirm if the slopes in the map are for each year or the 1979-2020 period. The values seem too low to correspond to the 42 years. Please confirm.

As per my previous revision, research on trends of maximum and minimum air temperatures and ETCCDI indices in Mainland China is not new since many research papers have been published on these subjects. The authors have replied to this. However, this should be clearly stated in the manuscript. What are the advantages of employing reanalysis data and seasonal trend analysis? What are the contributions of this study in comparison to previously published research articles? Do the results of this research follow the results of previously published papers although using different methodologies? You answer all these questions in the cover letter to me; however, you should state all these answers in the introduction and methodology sections in the manuscript for clarity.
